# Human occupations at the Alpysbaev Cave (western Tian Shan): Bioarchaeological insights from the Iron Age burial cluster

Abay Namen[1,2*], Saine C. Hernandez Burgos[3⊙], Aristeidis Varis[2,4], Emily Coco[3], Rachel Kalisher[3], Emily Gaul[5], Guido Alberto Gnecchi-Ruscone[6], Maria A. Spyrou[5,6], Cosimo Posth[5,6], Susanne Lindauer[7], David Naumann[4], Scott A. Williams[3], Zhaken Taimagambetov[8], Radu Iovita[2,3*]

**1** Department of Sociology and Anthropology, School of Sciences and Humanities, Nazarbayev University, Astana, Kazakhstan, **2** Early Prehistory and Quaternary Ecology, Department of Geosciences, University of Tübingen, Tübingen, Germany, **3** Center for the Study of Human Origins, Department of Anthropology, New York University, New York, United States of America, **4** Institute for Archaeological Sciences, Department of Geosciences, University of Tübingen, Tübingen, Germany, **5** Archaeo- and Palaeogenetics, Institute for Archaeological Sciences, Department of Geosciences, University of Tübingen, Tübingen, Germany, **6** Senckenberg Centre for Human Evolution and Palaeoenvironment at the University of Tübingen, Tübingen, Germany, **7** Curt-Engelhorn-Centre for Archaeometry, Mannheim, Germany, **8** National Museum of the Republic of Kazakhstan, Astana, Kazakhstan

⊙ These authors contributed equally to this work.
* abay.namen@nu.edu.kz (AN); iovita@nyu.edu (RI)

## Abstract

For millennia, southern Kazakhstan has been at the center of population movements and cultural exchange, hosting numerous tribal unions and confederations. The social structures of the societies that formed these early states have been the subject of extensive research, interpreted primarily from burial structures and funerary rites. In a landscape dominated by kurgans, catacombs, and necropoles, little is known about the disposal of the dead in natural shelters like caves. In this paper, we present the initial results from the newly excavated site of Alpysbaev Cave located in Turkestan Province, southern Kazakhstan. Test excavations yielded several intersecting pits which contained disturbed adult and nonadult human remains (MNI=4) as well as ceramic sherds, lithics, and by-products of combustion features. We radiocarbon dated material from our five lithostratigraphic units, which come from at least three distinct use phases spanning the Neolithic to early medieval and Iron Age periods. While the earliest lithostratigraphic unit contained human cranial fragments and faunal remains, most skeletal remains come from the Iron Age. We then present an integrated bioarchaeological and genetic evaluation of these remains and show evidence for subsistence practices, physical labour and pathological lesions among our sample.

## 1 Introduction

The piedmonts of the Inner Asian Mountain Corridor (IAMC) acted as a natural corridor for the movement of various populations from the Palaeolithic to historic

**Data availability statement:** All relevant data are within the manuscript and its Supporting Information files.

**Funding:** This project has received funding from the European Research Council (ERC) under the European Union's Horizon 2020 research and innovation programme (grant agreement n° 714842; PALAEOSILKROAD project).

**Competing interests:** The authors have declared that no competing interests exist.

periods [1–4]. With the intensification of human occupation from the early and middle Holocene to the late medieval period, the northern foothills of the western Tian Shan (Kazakhstan) have been part of numerous early states, tribal unions, and chiefdoms [2,5,6]. A recent genetic study found that multiple populations from the south (Kangju) and east (Xiongnu) expanded to the Kazakh steppes during the Iron Age and subsequent periods [4]. The genetic models demonstrate the heterogeneity of the populations with high eastern Eurasian admixture or with gene flow from South Asia [4]. These population shifts resulted in cultural transitions, often appreciable in changing funerary practices. From the 1st century BC, burial goods such as pottery, jewelry, arrowheads, and swords, also become more common, possibly due to the interaction with foreign groups [4]. Burial styles shifted from simple pits without additional structures during the 3rd and 2nd centuries BC, to underground catacomb burials in the 2nd and 3rd centuries AD [5]. By the 3rd and 4th centuries AD, the people of southern Kazakhstan, particularly the Kangju, used kurgans (types of tumuli constructed over a grave), necropoles, stone structures, and other burial structures [7–10]. Through the lenses of bioarchaeology and genetics, the Central Asian Iron Age is therefore an important lynchpin for understanding the broader population dynamics and settlements of the region.

Traditionally, natural shelters such as caves and rockshelters have not been recognized as possible burial venues for western Tian Shan in the early first millennium AD. These natural shelters have mostly been associated with Palaeolithic hominins. However, recent excavations demonstrate that such shelters have also been extensively occupied by diverse human groups throughout the Holocene and extending into the historic period [11–14]. Additionally, despite the large-scale excavations of burial sites, we still lack detailed data about the actual people of southern Kazakhstan. Given that Central Asia and southern Kazakhstan were defined by diverse multicultural intersections, social, and frequent demographic changes, conducting bioarchaeological studies on human remains combined with genetic research provides an opportunity to explore demographic structures and lifestyles.

In this article, we present the results of the excavations at the newly discovered site at Alpysbaev Cave. We first overview the stratigraphy of the cave, including the finds and radiocarbon dates from each level. We then contextualize the human remains found in these layers through bioarchaeological and preliminary genetic studies on endogenous and pathogen ancient DNA. Ultimately, we integrate these data and offer insights into what we argue was Iron Age population lifestyle in the 3rd and 4th centuries AD.

## 2  Alpysbaev Cave and the regional setting

Alpysbaev Cave (initially recorded as Saryaigyr cave by the PALAEOSILKROAD project) is located in the north-western foothills of the Tian Shan mountains, Turkestan Province, southern Kazakhstan (Fig 1). Administratively, it is in the territory of the Sairam-Ögem national park at the absolute altitude of ca. 1700 m.a.s.l. It is a small triangular-shaped cave situated in the Saryaigyr Valley on the left bank of the eponymous river (Fig 2). The site was discovered by a prominent Kazakh archaeologist,

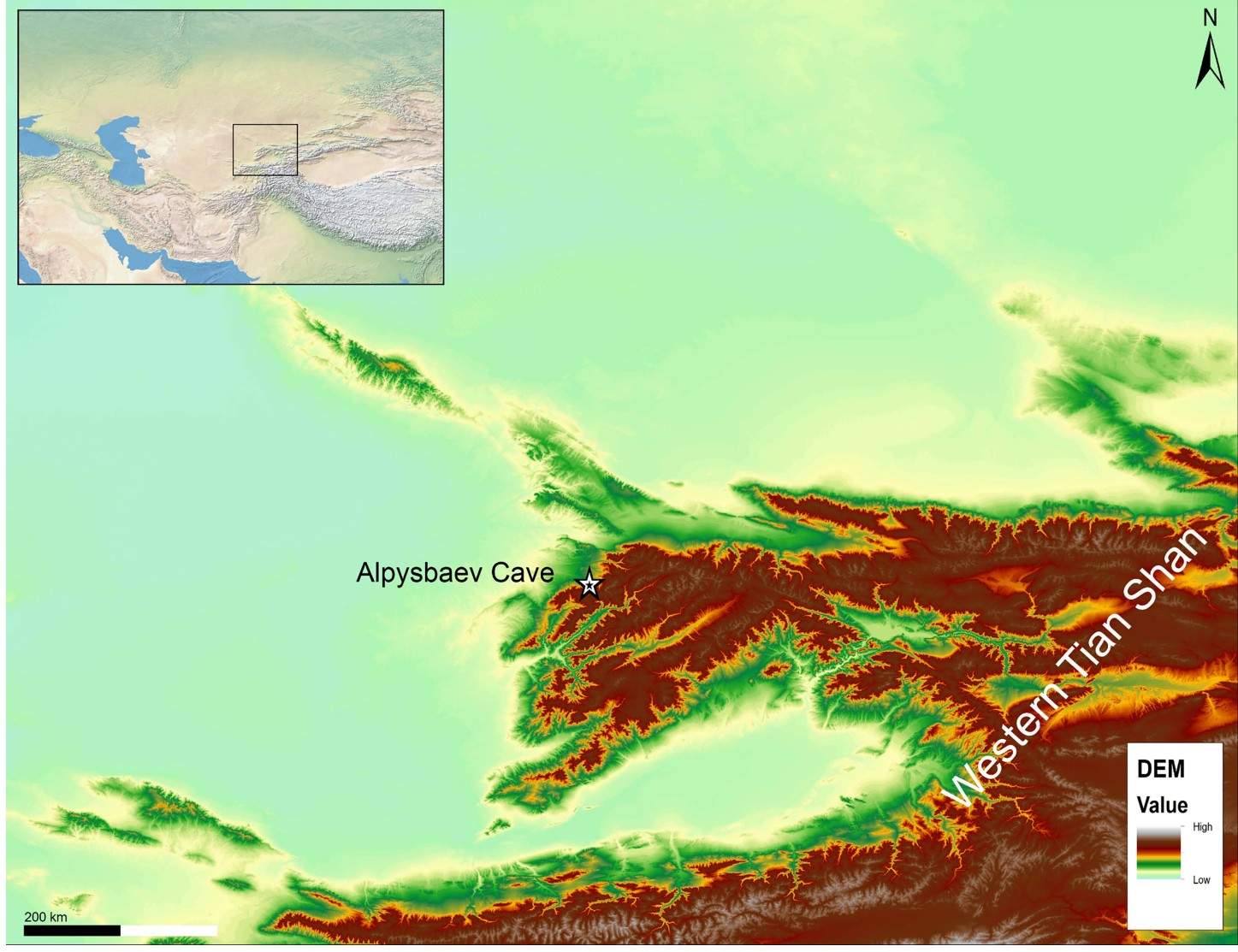

**Fig 1. The location of the Alpysbaev Cave is shown in relation to the local topography.** Data source: Global Administrative areas (GADM); vector and raster data from Natural Earth (www.naturalearthdata.com) and Shuttle Radar Topography Mission (SRTM) Version 4.

Khasan Alpysbaev in the 1960s, however, he did not conduct any archaeological excavations. Our team revisited the cave during an exploratory archaeological survey aimed at finding karstic caves and rockshelters using predictive modeling in 2019 [15]. Since our cave and rockshelter database accumulated several localities with the Saryaigyr name, we decided to name the cave after Khasan Alpysbaev. Initial geoarchaeological investigations with a penetrometer probe revealed a minimum sediment cover of about 80 cm, thus making it a promising location for excavation.

## 3 Materials and methods

### 3.1 Archaeological excavations

All necessary permits were obtained for our study, which complied with all relevant regulations. The study was conducted within the PALAEOSILKROAD project, and all field research was conducted under license n° 15008746 (12.05.2015)

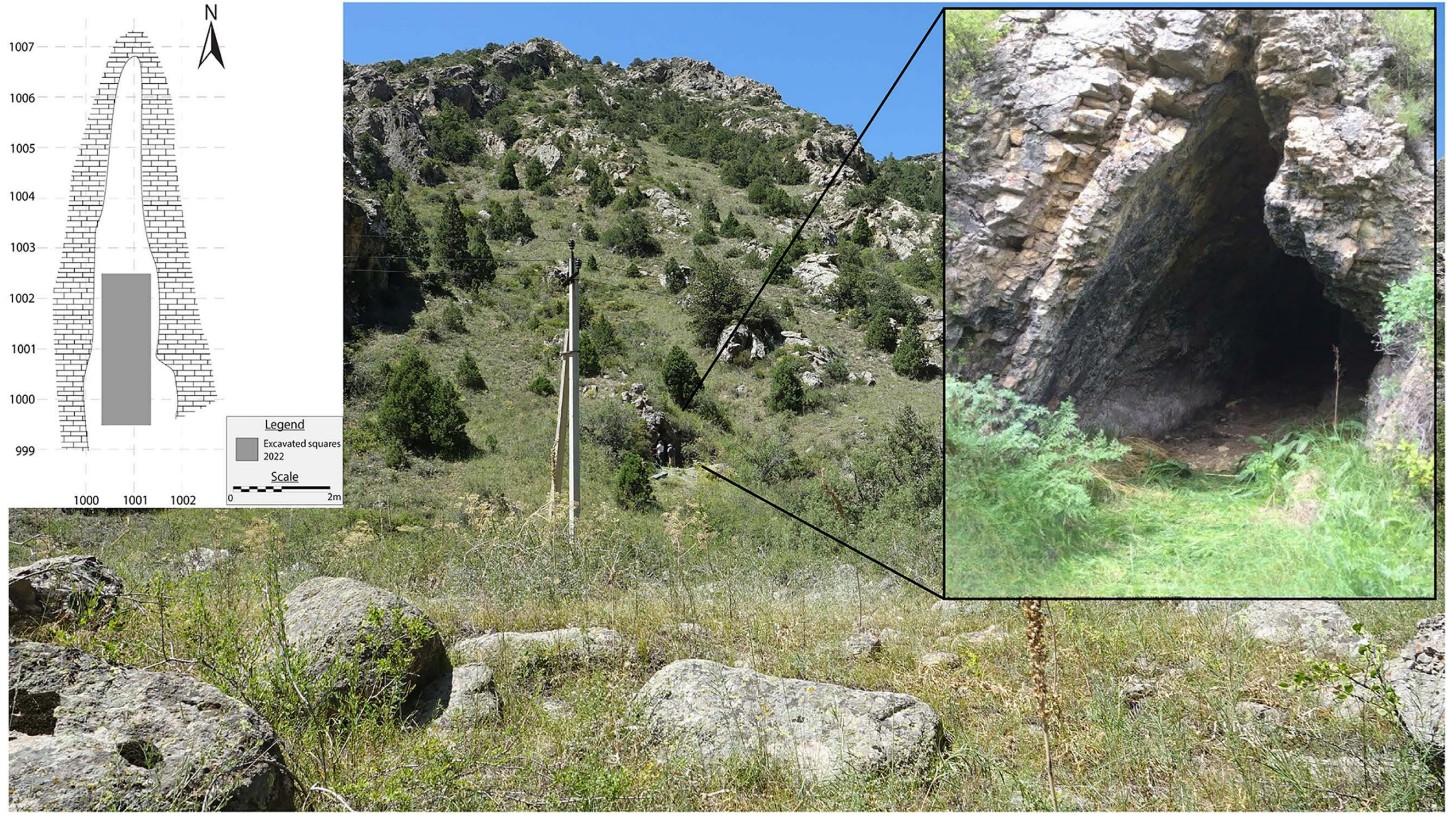

**Fig 2. View to the Alpysbaev Cave and the surrounding landscape.**

of the National Museum of the Republic of Kazakhstan, based on the collaboration protocol between the University of Tübingen and the National Museum. Our team conducted archaeological excavations at the site in 2022. A test pit measuring 3x1 m was opened at the cave entrance and extended into the interior. The excavations were conducted in a meter grid system, ensuring the correct documentation of archaeological materials found in the pit. The position of all artefacts and bones >1,5 cm was mapped with a total station. Excavated sediments were dry-sieved using 6 and 2 mm mesh. The lithostratigraphic units were described according to common geological fieldwork techniques and procedures [16]. Human skeletal materials were identified in the field by D.N.

### 3.2 Radiocarbon dating

We selected a total of eight well-contextualized samples consisting of charcoal fragments, animal, and human bones (Table 1) for C14 dating. Analysis was conducted at the Curt-Engelhorn Centre for Archaeometry, Mannheim (Germany). Bone collagen was extracted as described in Lindauer et al. [17], and charcoal samples received an ABA (acid-base-acid with diluted HCl and NaOH) treatment to remove contamination. The samples were combusted, graphitised, and measured as described in Lindauer et al. [17] and calibrated using the IntCal20 dataset [18] with Oxcal 4.4 [19].

### 3.3 Bioarchaeological and genetic analysis

All human remains were assigned numbers in the field, and analyzed and 3D scanned by SCHB. They are currently stored at Al Farabi Kazakh National University, in Almaty, Kazakhstan. Age was estimated from long bone size and

**Table 1. Radiocarbon dates of the Alpysbaev Cave were calibrated using Oxcal 4.4 with the IntCal20 dataset. It must be noted that the δ13C values are measured with an AMS system, not with an IRMS for stable isotope, and hence cannot be used for interpretation. They are used for the correction of fractionation occurring during measurement.**

| Lab Nr (MAMS) | Sample name | Level | ¹⁴C Age [yr BP] | ± | Cal 1-sigma 68% | Cal 2-sigma 95% | C:N | C [%] | Collagen [%] | Material |
|---|---|---|---|---|---|---|---|---|---|---|
| 60497 | TPIT1–50 | LU3 | 1135 | 17 | cal AD 888–972 | cal AD 776–986 | | 59.5 | | Charcoal |
| 60498 | TPIT1–283 | LU3 | 1124 | 17 | cal AD 893–974 | cal AD 887–987 | | 60.0 | | Charcoal |
| 60499 | TPIT1–350 | LU3 | 1103 | 17 | cal AD 899–990 | cal AD 892–994 | | 66.6 | | Charcoal |
| 60500 | TPIT1–503 | LU3 | 1140 | 18 | cal AD 885–972 | cal AD 774–980 | | 64.5 | | Charcoal |
| 60501 | TPIT1–524 | LU5 | 6316 | 22 | cal BC 5316−5221 | cal BC 5334−5215 | 3.2 | 37.9 | 8.4 | Animal bone |
| 61434 | TPIT1–330 | LU2 | 1776 | 20 | cal AD 244–325 | cal AD 234–340 | | 44.1 | | Human bone |
| 61435 | TPIT1–426 | LU3 | 1868 | 21 | cal AD 130–213 | cal AD 125–230 | | 44.1 | | Human bone |
| 61436 | TPIT1–389 | LU3 | 1828 | 20 | cal AD 207–245 | cal AD 130–313 | | 43.1 | | Human bone |

epiphyseal fusion, following the methods of Scheuer and Black [20] and Baker et al. [21]. Sex estimation was possible in one instance, on a complete right ossa coxae (TBIT-51), using pubic morphology of Klales et al. [22] and greater sciatic notch width of Walker [23]. Age and fragment location were used to create a minimum number of individuals (MNI) [24,25]. Bone abnormalities, pathologies, and postmortem damage were likewise identified and described when relevant. Further, entheseal changes to bone in response to musculoskeletal stress were assessed through the presence of osteophytic reactions and muscle scar rugosity.

In addition to bioarchaeological analysis, all teeth recovered from the site were sent to Archaeo- and Palaeogenetics lab of the University of Tübingen for ancient DNA analysis (aDNA) in hopes of identifying both endogenous and pathogen DNA. Each was sectioned at the cemento-enamel junction and a dental drill was used to remove ca. 40–70 mg of powderised dentine from the inside of the pulp chamber. DNA was extracted from the powderised material, following an established aDNA extraction protocol specialised for retrieving short and damaged DNA fragments [26]. Subsequently, 25 µl of DNA extract were converted into double-stranded DNA libraries using a protocol that included a partial uracil–DNA–glycosylase (half-UDG) treatment [27]. Shotgun high-throughput sequencing (SGS) was then performed on the prepared DNA libraries to a depth of ~8–10 million reads per genomic library. For assessment of human DNA content, SGS reads were aligned against the hg19 reference genome using the Burrows-Wheeler Algorithm (BWA) as part of the EAGER pipeline (v1.92.38, [28]). The SGS reads mapped to the human reference genome were used to infer the genetic sex of the six teeth using the sex determination method described in Skoguland et al. [29].

The SGS reads were then subjected to a metagenomic screening using the MALT/HOPs pipeline [30], which allows for the identification of ancient microbial DNA. Identified pathogens were validated through qualitative assessment, considering the distribution of reads mapping to each respective reference sequence, the mapping of reads to non-repetitive regions, and read specificity to species of interest.

## 4 Results

### 4.1 Stratigraphy

The excavated sequence currently consists of 5 lithostratigraphic units (LUs), with the deepest part of the test excavation reaching a depth of 1.80 m (Fig 3). This deepest layer, LU5, was excavated in a small area at the northwest corner of the trench (Figs 3 and 4). It is a dark brown sediment rich in pebble-sized angular gravels, animal bones, and human cranial fragments (Fig 4A). The boundary between LUs 5 and 4 is unclear due to the confined excavation area. LU4 is a yellowish-brown sediment with few finds in the northern half of the trench. It has a sandy loam, non-plastic and loose texture with a moderate amount of pebble and cobble-sized gravels. A smooth and abrupt contact marks the transition

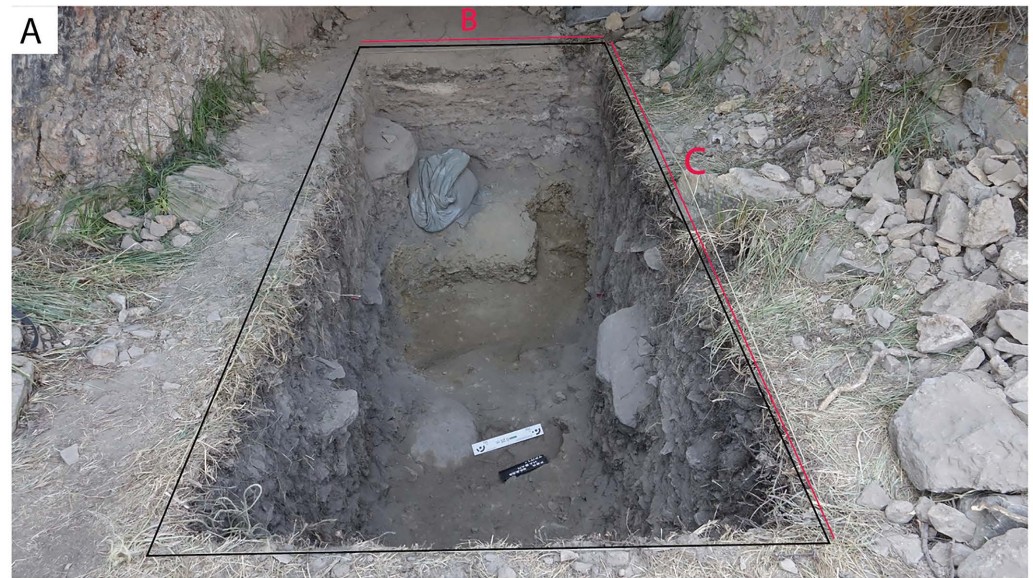

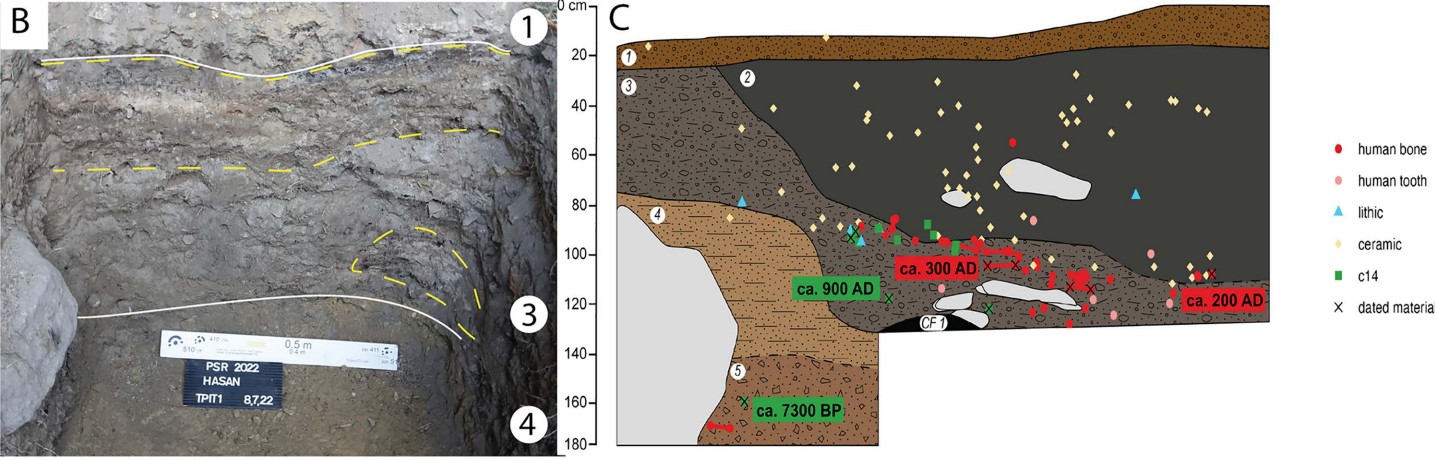

**Fig 3. Overview of the 2022 excavations. A)** Overview of the test-trench towards the end of the excavation. The oldest date is obtained from an animal bone in LU5, where human cranial fragments were found. In red colors: B – northern profile, and C – eastern profile. **B)** Detail of the northern profile demonstrating the sharp and smooth boundaries (continuous white lines) between the more geogenic LU4, the more heterogeneous LU3 with *fumier* layers or lenses (dashed yellow lines), and LU1. The discontinuity between the *fumier* deposits might result from bioturbation. **C)** The stratigraphy sketch on the eastern profile demonstrating the distribution of archaeological finds and the radiocarbon dates.

from LU4 to LU3 at the north profile (Fig 3B). However, the contact between LU4 and LU3 becomes irregular towards the south, suggesting that LU4 is cut by LU3 (Fig 3C). The irregular contact between LU4 and LU3 and the presence of thicker LU4 deposits towards the northern part of the trench suggest the removal of LU4 sediments from pit-digging activities occurring during the formation of LU3. The stratigraphy becomes more complex towards the middle of the trench. Under an accumulation of boulders, a dark brown deposit rich in comminuted charcoal fragments was exposed (Fig 3B). It has a sandy loam, friable, and non-plastic texture with a low content of small angular gravels, but with frequent granule-sized lumps of orange and yellowish sandy clay. This deposit was unique in Alpysbaev Cave and was excavated as a separate feature, probably associated with combustion activities and recorded as a combustion feature

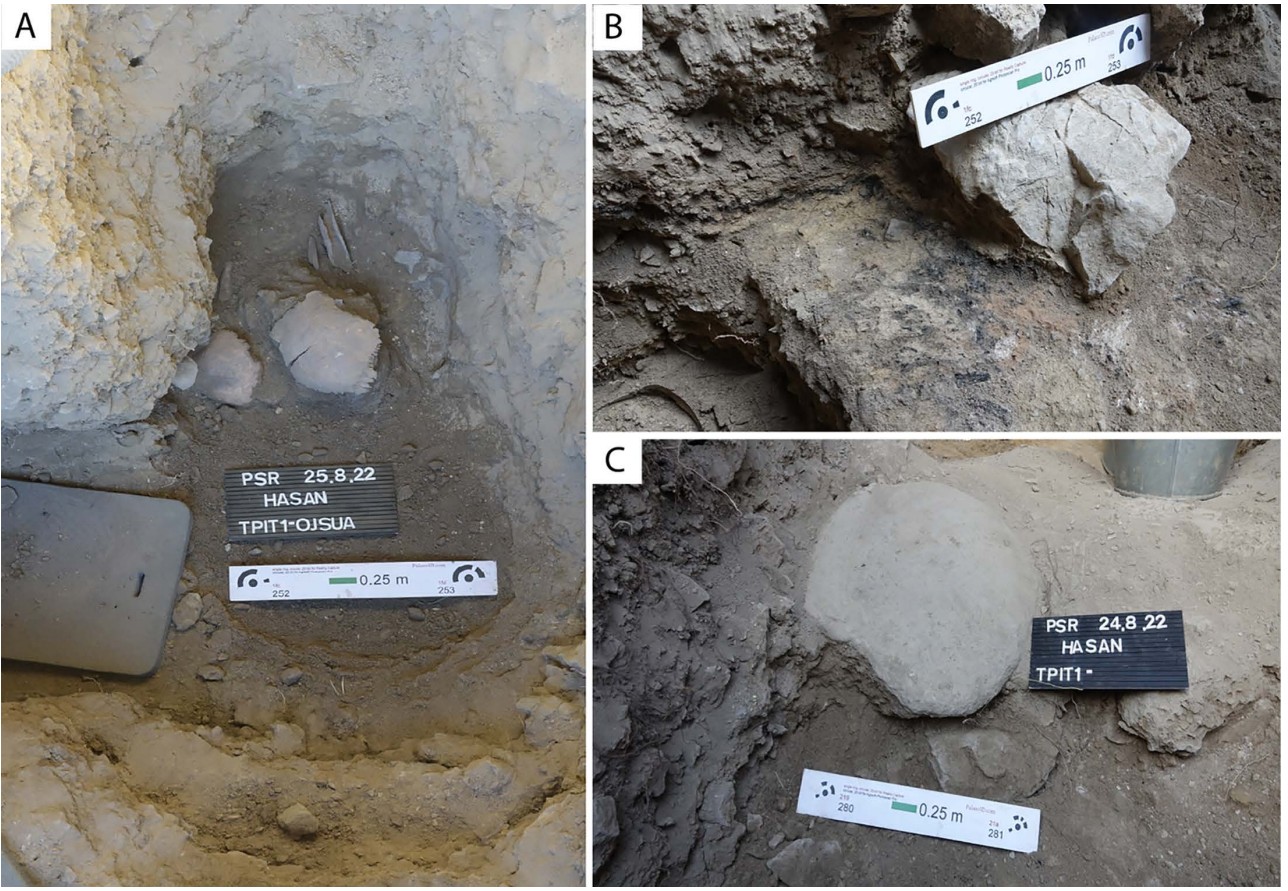

**Fig 4. Archaeological contexts. A)** Human cranial fragments and animal bones found in LU5. **B)** Charcoal-rich sediments excavated as combustion feature 1 (CF1). **C)** River cobble manuport, possibly used as a heat retainer.

(CF1) (Fig 4B). CF1 could be a localised lens or a more extensive layer, but its spatial extent is uncertain due to limited excavation. LU3 overlies both LU4 and CF1. It is a greyish-brown deposit containing interbedded layers and lenses of varying colours that have a loose, silty texture and are rich in organics, ashes, or more granular material. Archaeological deposits with these lithological characteristics could represent *fumier* deposits, which are often found at Holocene cave sequences and indicate the stabling of animals [31]. In Kazakhstan, the use of caves for stabling during the Holocene is also known from Aqtogai cave [14]. LU3 is bioturbated and contains a moderate amount of gravel ranging from pebbles to boulders. It has the densest concentration of human bone, and contains lithics and pottery sherds. LU2 is a highly bioturbated and very loose granular sediment covering the majority of the upper part of the sequence at Alpys-baev Cave. The boundary between LU2 and LU3 is unclear because of bioturbation, but LU2 seems to intersect the *fumier* layers identified on LU3. Therefore, we hypothesise that LU2 represents a more recent pit-digging event. This is possibly due to human activities during the Middle Ages, as the presence of medieval pottery from this layer suggests. Isolated human bones were also found in LU2. Finally, the topmost layer, LU1, is a thin dark brown surface layer rich in humic material, modern organic matter, and roots.

Overall, the archaeological sequence towards the back of the cave remains undisturbed, with a clear division of litho-stratigraphic units. The majority of the finds derive from the area closer to the cave entrance, where a series of pit-digging activities has been recorded. Finds near the cave mouth include ceramic sherds, stone tools, manuports, faunal, and

human remains (Fig 4). Due to time constraints, we opted to stop the excavations of the intact middle Holocene unit and return in the future to conduct a longer field season at the cave.

## 4.2 Chronology

Of the five LUs, we were able to retrieve dates from three layers (LU2, LU3, LU5) (Table 1). The radiocarbon dates from LU3 fall into two different clusters (Table 1). Chronometric dates of the human remains correspond to the Iron Age (125–340 cal AD), whereas charcoal fragments from the same unit are dated to the early Medieval period (774–994 cal AD). The temporal disparity between various samples from the same unit most likely resulted from the mixing of occupational horizons from the top layers. The animal bone sampled from the undisturbed LU5 (and associated with the human cranial fragments) yielded a date of 6316±22 uncal BP (calibrated to 7.3 ky BP), which corresponds to the early Neolithic period in the region.

## 4.3 Human remains

Test excavations revealed several pits in LUs 2 and 3 that intersect one another and contain disturbed human remains. No bones were found in articulation, and their stratigraphic position is illustrated in Fig 3C. Given that the remains are not in their anatomical position; it is possible that they were once primary inhumations, that were disturbed by pits dug during the early Medieval period (800–900 CE).

In addition to the human remains from LUs 2 and 3, two human cranial fragments and several faunal remains were found in undisturbed LU5 (Fig 5). This unit was only exposed towards the interior part of the cave, and excavations were paused due to time constraints. A radiocarbon date of the animal bone associated with the human crania (MAMS-60501) corresponds to the Neolithic period (7.3 ky cal. BP), suggesting human remains have been sporadically deposited at this site for at least six millennia.

**4.3.1 Bioarchaeological analysis.** The excavated sample consists of 41 human bones and teeth (Table 2). The preservation of the skeletal materials varies from well-preserved to poor and highly fragmented. Besides teeth, the skeletal remains in this sample include only postcrania: vertebrae, clavicles, ribs, os coxae, scapulae, phalanges, and a tarsal. Out of this sample, only one bone was unidentified. While all but six skeletal elements belonged to adults (see Table 2), interestingly, only small-to-medium sized skeletal elements were recovered. The largest skeletal elements found were the preserved ossa coxae (TPIT1–51) and a humerus (TPIT1–40). Detailed descriptions and photos of each of these skeletal elements are available in the supplementary material (Text S1 in S1 File).

A minimum number of 4 individuals (MNI) is represented in Alpysbaev Cave. Given the lack of disturbance in LU5, we do not believe the human cranium from this deepest layer (MNI = 1) corresponds with the skeletal material from the more heavily disturbed later layers. All remaining skeletal elements come from LUs 2 and 3. Because the radiocarbon dates indicate that the humans in these layers did not occupy the layer in which they are deposited, we consider all remains from LUs 2 and 3 together as a closed assemblage. The adult skeletal remains represent at least two individuals, evidenced by two right clavicles (TPIT1–44 and TPIT1–319) belonging to young adults of a similar age inferred from the lack of fused medial epiphyses (19–30 years). The third clavicle, a left (TPIT1–330), likely belongs to the same individual as TPIT1–44. As the medial epiphysis of the clavicle is the last epiphysis to fuse, all other skeletal elements would appear fully mature, meaning it is not possible to seriate the other bones as belonging to "young adults." All six teeth exhibited severe dental wear, with no repetitions in position, and theoretically may have belonged to one individual. However, genetic sex determination of the teeth suggests that they belonged to at least two individuals–one male and one female (see below). For the nonadult remains, it is possible that the lower thoracic and lumbar vertebrae (TPIT1–39 & 42) belong to the same child, not only because their centra are similar in size, but also because their development aligns with the fusion sequence of a child of roughly

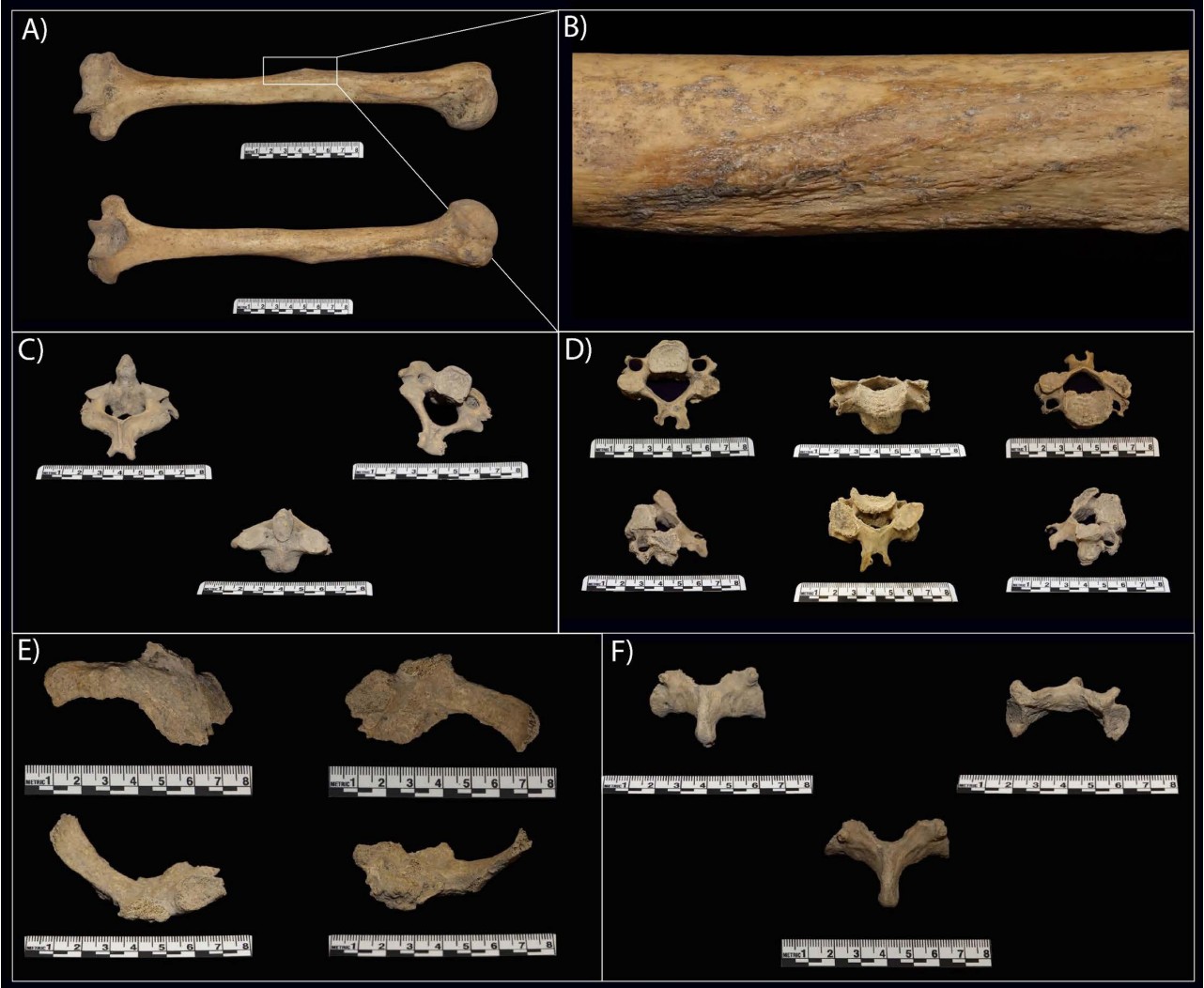

**Fig 5. Skeletal remains illustrating observed bone modifications: A) and B) demonstrate deltoid tuberosity of TPIT1–40. C)** Cervical vertebra (C2), TPIT1–523, shows the small osteophyte on the right portion of the bifurcated spinous process. **D)** Complete adult cervical vertebra (TPIT1–398) exhibiting evidence of degenerative joint disease, also known as osteoarthritis. **E)** A fragmented left rib of TPIT1–389 (anterior – top left, posterior – top right, superior – bottom left, inferior – bottom right) presenting significant bone remodeling as a result of bone fracture healing. **F)** Partial neural arch of TPIT1–178. Based on the size and orientation of the inferior articular facet, this specimen is a lumbar vertebra missing the centrum, pedicles, and superior articular facet. This specimen presents evidence of bilateral spondylolysis.

4–6 years [20]. It is also possible that the nonadult rib (TPIT-326) belonged to this child. It is likewise possible that the unfused ilium (TPIT1–491) belonged to this same child as the age estimate for its state of nonfusion suggests a 6–9 year range. Finally, one skeletal element may belong to a fetal individual, however its morphology is currently not diagnostic enough to determine the element with any certainty. Thus, a conservative MNI for this cave is 4, with a more liberal interpretation leading to an MNI of 5.

The sex estimation of the adult, right ossa coxae (TPIT1–51), suggests that the individual was male. This estimation would have been more accurate if more of the features used for the assessment of sex were intact, but the morphology of the greater sciatic notch [32] and the obturator foramen [33,34] are consistent with male specimens.

**Table 2. Inventory of human skeletal remains from Alpysbaev Cave. Asterisk denotes the sample was radiocarbon dated.**

| Arch ID (Lab ID) | Side+Element | Age Category | Age Range | Notes |
|---|---|---|---|---|
| TPIT1–36 | Right Rib 12 | Adult | – | |
| TPIT1–37 | Unsided scapula | Adult | – | Lateral border |
| TPIT1–39 | Lower lumbar vertebra | Nonadult | 5+ years | Unfused apophyses |
| TPIT1–40 | Right humerus | Adult | – | Pronounced deltoid tuberosity |
| TPIT1–43 | Lower thoracic vertebra | Nonadult | 4-6 years | |
| TPIT1–44 | Right clavicle | Young adult | 19-30 years | Sternal epiphysis unfused; acromial epiphysis fused |
| TPIT1–49 | Right typical rib | Adult | – | Healed bony callus on caudal border |
| TPIT1–51 | Right ossa coxae | Adult | – | Male sex estimate; age estimate not possible due to damaged pubic symphysis |
| TPIT1–229 (TPIT001.A) | Upper left canine? | Adult | – | |
| TPIT1–318 | Left typical rib | Adult | – | |
| TPIT1–319 | Right clavicle | Young adult | < 19–30 years | Sternal epiphysis unfused; unknown acromial epiphysis fusion; pronounced deltoid muscle attachment, possibly belongs with TPIT1–40 |
| TPIT1–326 | Left typical rib | Nonadult | < 17–25 years | Billowed sternal end; unfused tubercle and head |
| TPIT1–330* | Left clavicle | Young adult | 19-30 years | Sternal epiphysis unfused; possibly belongs with TPIT1–44 |
| TPIT1–332 | Left typical rib | Adult | – | |
| TPIT1–336 | Left typical rib | Adult | – | |
| TPIT1–341 | Unsided scapula | Adult | – | |
| TPIT1–356 | Right cuboid | Adult | – | |
| TPIT1–359 (TPIT002.A) | Lower left P3? | Adult | – | Caries on distal CEJ |
| TPIT1–361 (TPIT003.A) | Lower left M1 | Adult | – | Moderate dental wear |
| TPIT1–376 | Left typical rib | Adult | – | |
| TPIT1–384 | Potential fetal human bone | Fetal | – | Woven bone indicates fetal individual, but morphology is unclear |
| TPIT1–389* | Right typical rib | Adult | – | Healed bony callus |
| TPIT1–398 | Typical cervical vertebra | Adult | – | Osteoarthritic lipping on left superior articular facet |
| TPIT1–405 | Left scapula | Adult | – | |
| TPIT1–406 | Thoracic vertebra, spinous process | | – | |
| TPIT1–410 | Proximal foot phalanx, ray 2–5 | Adult | – | |
| TPIT1–425 | Proximal hand phalanx, ray 2–5 | Adult | – | |
| TPIT1–426* | Right typical rib | Adult | – | |
| TPIT1–450 (TPIT004.A) | Lower left P4? | Adult | – | |
| TPIT1–463 (TPIT005.A) | UID molar | Adult | – | Half of tooth is broken, root missing antemortem (likely secondary to infection) but both the mesial and distal surfaces are eburnated. |
| TPIT1–473 | Right rib 1 | Nonadult | < 17 years | |
| TPIT1–474 (TPIT006.A) | Lower right P3 | Adult | – | |
| TPIT1–484 | Intermediate hand phalanx | Adult | – | |
| TPIT1–489 | Proximal hand phalanx, ray 2–5 | Adult | – | Carnivore gnawing on distal head |
| TPIT1–491 | Right ilium | Nonadult | 6-9 years | Ilium fully unfused at acetabular border |
| TPIT1–492 | Intermediate hand phalanx | Adult | – | |
| TPIT1–495 | Intermediate and distal hand phalanx (set), ray 2–5 | Adult | – | |

*(Continued)*

**Table 2.** (Continued)

| Arch ID (Lab ID) | Side+Element | Age Category | Age Range | Notes |
|---|---|---|---|---|
| TPIT1–523 | C2 cervical vertebra (axis) | Adult | – | |
| TPIT1–532 | Right typical rib | Adult | – | |
| TPIT1–NoContext | Lower thoracic vertebra (T11) | Adult | – | |

**4.3.3 Musculoskeletal stress markers.** Evidence of musculoskeletal stress markers (MSMs) is evident in three specimens (TPIT1–523, TPIT1–398, and TPIT1–40) (Fig 5). The study of musculoskeletal stress markers and associated muscle functions can be used to infer the activity patterns and lifestyles of past human populations [35,36].

The osteophyte present on the right portion of the bifurcated spinous process of the adult second cervical vertebra, TPIT1–523, may be an indication of biomechanical strain at the muscle attachment, or may also be age-related (Fig 5C). The adult typical cervical vertebra, TPIT1–398, also presents evidence of robusticity of the spinous process which also indicates hypertrophy at this site of muscle attachment. The muscles that interact with the spinous processes of the typical cervical vertebrae and C2 are: spinalis muscles, interspinales muscles, semispinalis muscles, multifidus muscles, and rotatores muscles [37]. These muscles are involved in extension and rotation of the neck and spine [37,38].

Entheseal change was also present on the adult humerus, TPIT1–40 (Fig 5A and 5B). There, the rugosity of the deltoid tuberosity is well-defined and raised. This indicates hypertrophy at this site of muscle attachment most likely triggered by intense and frequent muscle activity. The function of the deltoid muscle is to abduct the shoulder, flexion, and extension. Interestingly, the two right young adult clavicles also exhibit entheseal changes at the location where the deltoid inserts, either suggesting that the humerus belonged to one of those two individuals or that the type of activity that heavily relies on the use of this muscle was common among the group.

**4.3.4 aDNA preservation.** Genetic analysis revealed human DNA across all six tooth samples to range from 75.48% −0.80% suggesting overall high preservation of genetic material at the site (summarised in Table 3 below). The authenticity of mapped reads was tested through the evaluation of patterns of deamination (see Fig S38 in S1 File). As hydrolytic damage occurs on DNA molecules over time, its presence can be considered indicative of aDNA [39,40]. Two samples were found to have deamination rates that are considered low; both TPIT1–359 and TPIT1–474 had damage rates of 4.37% and 4.99%, respectively. However, given the context, this is likely consistent with the excellent level of preservation observed across the sample set.

**Table 3. Selected statistics from the human screening of generated data, performed using the EAGER pipeline. Reads were filtered with Clip&Merge and mapped to the hg19 human reference using the BWA algorithm (parameters -n 0.01 -l 99999 -q 30), and duplicate removal was performed using DeDup. Filtering steps include the removal of reads which do not meet a minimum mapping quality score, and of PCR duplicates.**

| Arch ID | Lab ID | Raw sequenced reads | Mapped human reads | Mapped human reads post-filtering | Endogenous DNA (%) | Damage rate (%) | Average fragment length |
|---|---|---|---|---|---|---|---|
| TPIT1–229 | TPIT001 | 9,435,033 | 233,637 | 141,672 | 2.65 | 7.89 | 43.97 |
| TPIT1–359 | TPIT002 | 9,933,685 | 7,124,020 | 4,765,986 | 75.482 | 4.37 | 54.67 |
| TPIT1–361 | TPIT003 | 8,307,756 | 73,515 | 49,941 | 0.974 | 5.32 | 56.04 |
| TPIT1–450 | TPIT004 | 8,142,006 | 403,785 | 251,121 | 5.292 | 6.39 | 45.89 |
| TPIT1–463 | TPIT005 | 8,636,527 | 93,015 | 52,152 | 1.169 | 14.3 | 38.18 |
| TPIT1–474 | TPIT006 | 9,371,903 | 71,446 | 47,959 | 0.798 | 4.99 | 57.92 |

**4.3.5 Genetic sex.** All six teeth had enough reads mapping on the X and Y chromosomes to estimate genetic sex resulting in three teeth classified as female and three as male (Table 4).

**4.3.6 Palaeopathology and aDNA pathogen screening.** Three skeletal elements (TPIT1–178, TPIT1–398, and TPIT1–389) exhibit pathologies. A partial neural arch from a lumbar vertebra (TPIT1–178) is missing the centrum, pedicles, and superior articular facets (Fig 5F). This is consistent with bilateral spondylolysis, with evidence of healing of the stress fracture at the pars interarticularis [41].

A complete adult typical cervical vertebra, TPIT1–398, (Fig 5D) exhibits pathological markers that are mainly isolated to the left half of the bone. The left superior articular facet exhibits evidence of degenerative joint disease (DJD), or osteoarthritis. The surface of the left superior articular facet also presents evidence of porous degeneration, a feature that is not present on the right superior articular facet.

The final element that exhibits a pathological condition is TPIT1–389, a fragmented right typical rib (Fig 5E). This rib presents evidence of significant bone remodeling due to a healing bone fracture. The bony callus is still present and shows evidence of secondary infection, potentially osteomyelitis based on the presence of cloacae around the affected site. The size of the bony callus also indicates that the ribs superior and inferior to this one were affected and united by a bony ridge. Based on the location, it can be deduced that the fracture occurred at the costal angle, near the vertebral end (back). The degenerative joint changes of the rib head also suggest that the thoracic vertebrae were affected.

In addition to macroscopic observation of pathologies, metagenomic analysis of the teeth identified multiple oral pathogens, including *Parvimonas micra, Treponema denticola, Streptococcus mutans,* and *Campylobacter rectus* (Fig 6). A separate screening procedure for viruses was similarly carried out with MALT, however, in this case, none were identified.

## 4.4 Other archaeological remains

**4.4.1 Ceramics.** The ceramic assemblage consists of 374 fragments including piece-provenienced sherds and small fragments retrieved from bucket sieves. These finds were unearthed from LUs 1, 2, and 3 (see Fig 3). The vast majority of them derive from the disturbed LU2. Only one sherd was found towards the top surface of LU4, which is probably intrusive from LU3. Surface features (i.e., the trimming lines) in the exterior and interior of the ceramic fragments suggest the wheel-throwing production technique. Almost none of the ceramic fragments have traces of decoration. However, two sherds have glazed coating and are decorated with green-colored pigments (Fig 7). The wheel-thrown ceramic production and the radiocarbon dates from the disturbed LUs 2 and 3 suggest that these cultural layers were formed during the Medieval period.

**4.4.2 Lithic assemblage.** The co-occurrence of medieval pottery with stone tools indicates the disturbed nature of these cultural layers. The stone artefacts were found in association with ceramic fragments and human remains throughout the disturbed units LU2 and 3 (see Fig 3C). The lithic assemblage consists of 22 knapped stone artefacts, of which eleven

**Table 4. Genetic sex determination results performed with the "ry_compute" script that performs probabilistic assignments of the biological sex based on the ratio of the number of sequences mapping to the sex chromosomes (X and Y) over the other 22 autosomes. For sex assignment, male = XY, female = XX.**

| Arch ID | Lab ID | Tooth* | Nseqs | NchrY+NchrX | NchrY | R_y | SE | 95% CI | Sex Assignment |
|---------|--------|--------|-------|-------------|-------|-----|-----|--------|----------------|
| TPIT1–229 | TPIT001 | LC$^1$ | 141672 | 6135 | 62 | 0.0101 | 0.0013 | 0.0076-0.0126 | XX |
| TPIT1–359 | TPIT002 | LP$_3$ | 4765986 | 232302 | 1773 | 0.0076 | 0.0002 | 0.0073-0.008 | XX |
| TPIT1–361 | TPIT003 | LM$_1$ | 49941 | 1399 | 129 | 0.0922 | 0.0077 | 0.077-0.1074 | XY |
| TPIT1–450 | TPIT004 | LP$_4$ | 251121 | 6643 | 568 | 0.0855 | 0.0034 | 0.0788-0.0922 | XY |
| TPIT1–463 | TPIT005 | UID Molar | 52152 | 1314 | 117 | 0.089 | 0.0079 | 0.0736-0.1044 | consistent with XY but not XX |
| TPIT1–474 | TPIT006 | RP$_3$ | 47959 | 2206 | 15 | 0.0068 | 0.0017 | 0.0034-0.0102 | XX |

\* L = left, R = right, UID = unidentifiable, subscript = lower tooth, superscript = upper tooth

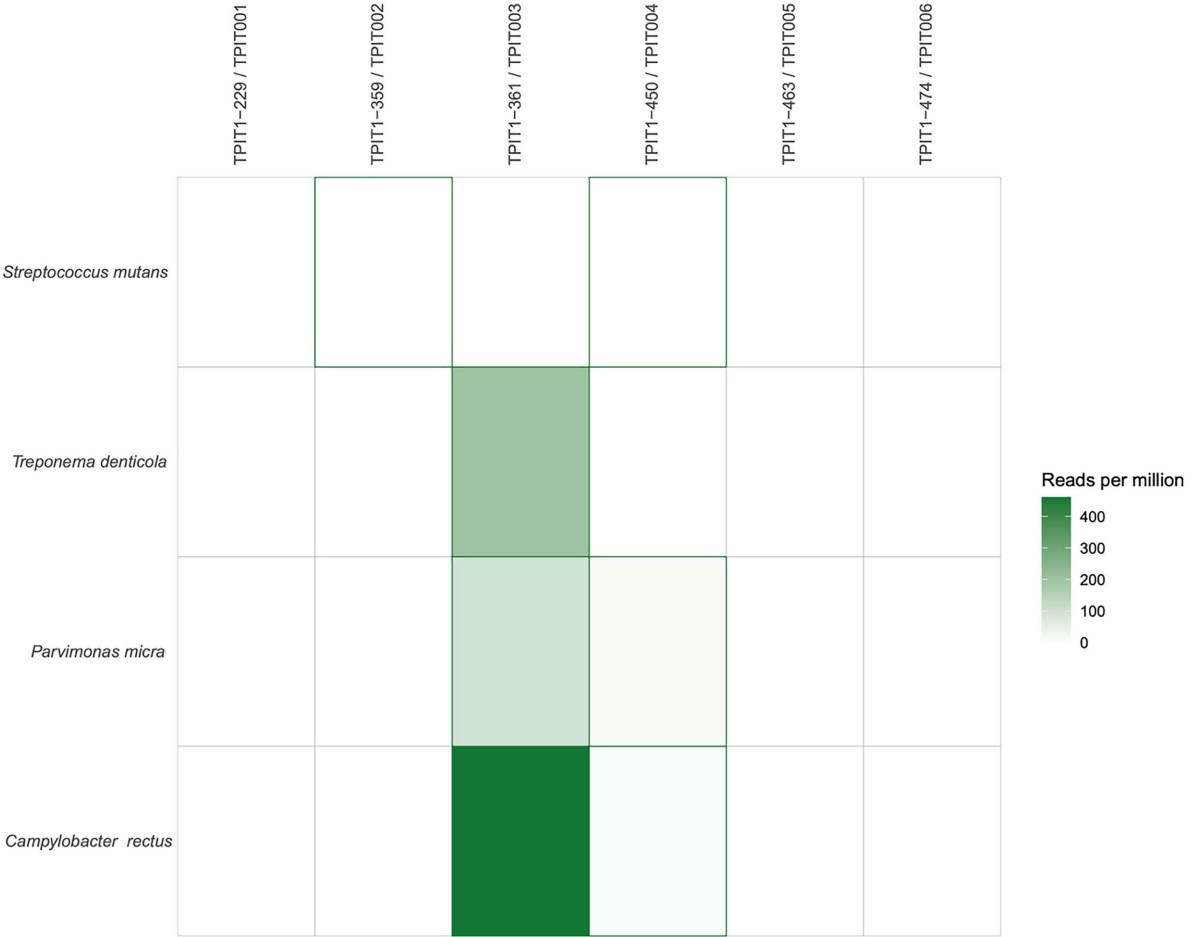

**Fig 6. Number of reads assigned to detected oral pathogens.** Reads were assigned using MALT (MEGAN Alignment Tool; [42,43]), which employs the naïve Lowest Common Ancestor (LCA) algorithm to align reads against a database of microbial reference genomes. Assigned reads were then further evaluated usings HOPs [30], which considers the edit distance distribution of reads, as well as patterns of deamination. Only species for which reads were manually validated through further qualitative assessment of read distribution, complexity and specificity are depicted here. Values are given as the number of reads attributed to a specific pathogen, for every million reads sequenced.

are micro-blades, and seven are flakes and flake fragments (Fig 8). Preliminary observations allow us to conclude that the primary technology was mainly focused on the production of bladelets and micro-blades from wedge-shaped and/or prismatic cores. The tool kit consists of four retouched micro-blade fragments and one complete endscraper. Among the micro-tools, one longitudinal edge retouch dominates, whereas only one micro-blade is retouched bifacially. The length of micro-tools ranges from 15 to 21 mm, the width is 3–5 mm. These artefacts could be preliminarily interpreted as insets for composite tools.

The rest of the assemblage is represented by small debitage. The presence of small chips might be an indication of knapping or retouching directly in the cave. The lithics are knapped on various raw materials including chert and shale. A preliminary survey of the surrounding landscape did not yield raw materials macroscopically similar to those of knapped tools. However, our previous work on the distribution of knappable raw materials demonstrates the occurrence of chert in the north-western foothills of Tian Shan [44,45].

Stone tool technology involving bladelets persisted in this region at least till 4000 years ago, well into the Bronze Age [13]. Therefore, bladelets are not synchronous with the medieval or even Iron Age deposits that characterise LUs 2 and 3.

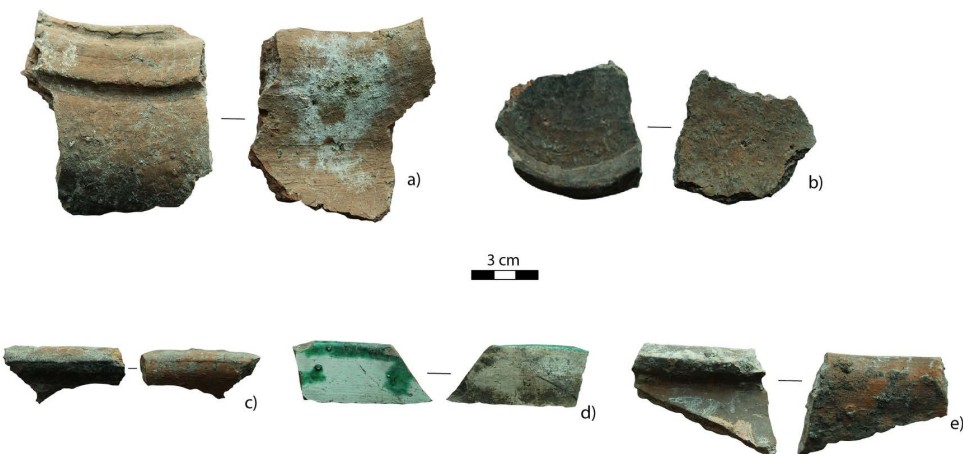

**Fig 7. Fragments of medieval utilitarian ceramics: a, c, and e) rim fragments, b) a fragment of a darkened ceramic base demonstrating that it has been exposed to fire, and d) a glazed ceramic sherd.**

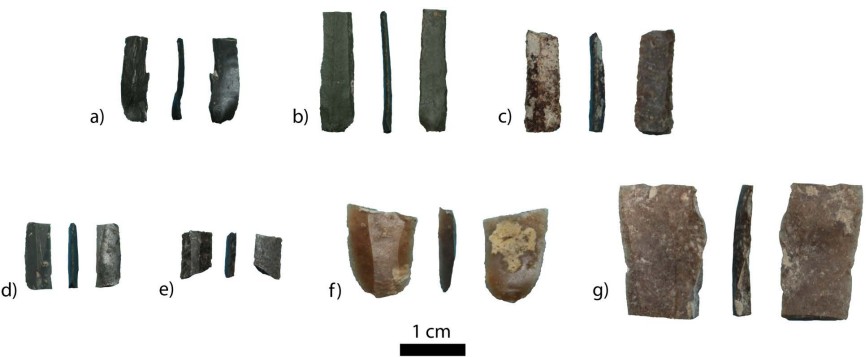

**Fig 8. Illustration of selected bladelets from Alpysbaev Cave.**

It is logical to hypothesise that the bladelet assemblage has been incorporated into LUs 2 and 3 by the same pit-digging activities that formed these deposits. The lithics could have been originally deposited in LU4, which is cut by LU3 at the southern part of the trench and only partially excavated at the northern part of the trench. The age of LU4 is still unknown given the absence of cultural materials or radiocarbon dates, the lack of which could be attributed to the limited excavation area. In the same notion, we cannot exclude the possibility that the bladelets could even be provenanced from LU5, where we have a documented presence of human bone fragments and faunal remains dated to the Neolithic. This correlation would assume that the pit-digging activities of LU3 intersected not only LU4 but also LU5 in the unexcavated southern part of our excavation trench. Further excavations in Alpysbaev Cave are necessary to shed light on the origins of the bladelet assemblage, whose presence, however, makes the preservation of early or middle Holocene occupation layers more likely.

## 5 Discussion

### 5.1 Human occupation at Alpysbaev Cave

Alpysbaev Cave provides a unique record of human occupation in the western Tian Shan (Kazakhstan), Central Asia, evidenced through ceramic, lithic and osteological remains. The excavated materials combined with the radiocarbon

dating allow us to propose at least three occupational phases: (1) early Medieval, (2) Iron Age, and (3) the early Neolithic. The most recent occupation occurred during the early Medieval period (ca. 900 AD). A large number of ceramic sherds, combustion features with charcoal fragments, and several absolute radiocarbon dates obtained from charcoal samples support this conclusion. The second occupational phase, based on the radiocarbon dating of human remains, falls within the boundaries of 120–340 AD. This chronology coincides with the Iron Age of the region. We hypothesise that at least four adult and nonadult individuals belonging to this period were buried in the cave. This is the first documented evidence of a cave burial in southern Kazakhstan. Based on the redeposited nature of the assemblage, we propose two possible explanations for why human remains are found in mixed contexts. First, it is possible that the human remains were disturbed shortly after the burial. Archaeological excavations of sites associated with the Iron Age in the area reveal evidence of early instances of theft of burial goods (for example see [5]). However, the second and more likely scenario is that the early Medieval population disturbed the earlier Iron Age burials when digging the pits necessary to build fireplaces and maybe store products underground in ceramic vessels. This is supported by numerous fragments of medieval ceramics found in association with human remains and lithic artefacts (Fig 8). This might also explain why we recovered few large skeletal elements, as these would have been the easiest to pick up and redeposit elsewhere. As a result, the smaller skeletal elements were left behind, leading to their recovery by our team millennia later.

Faunal and human remains combined with the absolute date of 7300 cal BP (MAMS-60501–6316±22 uncal BP), point to the oldest phase of occupation during the early Neolithic. Our excavations stopped once we reached the intact LU5 in the interior of the cave, where a high density of human and animal bones was found. It is plausible to assume that the unearthed cranial fragment is part of a Neolithic burial. However, additional excavations are necessary to support this interpretation. We can tentatively propose that the cultural layer with the early Neolithic artefacts may have been disturbed by the Iron Age populations to bury individuals or even by the early medieval activities. The presence of bladelets exhibiting typological similarities to early and middle Holocene industries, alongside human remains and ceramic sherds emphasise this observation. A penetrometer probe showed that sediments continue for at least another 70 cm, providing ample opportunity for documenting the Neolithic burials and potentially finding even older occupations.

There are only a few examples of early Neolithic burial practices across Kazakhstan. The recently discovered Koken site is currently the earliest directly dated early Neolithic burial (7.4 ka cal BP) accompanied by stone tools [46]. A similarly aged (ca. 7.7 ka cal BP) single find of a human bone in zooarchaeological museum collections from 1950s excavations of the now submerged Bukhtarma Cave in eastern Kazakhstan unfortunately lacks an archaeological context [47]. In southern Kazakhstan, excavation at the Qaraungir rockshelter, located in the north-eastern part of the Lesser Qaratau range, documented early instances of a Mesolithic burial [48]. Although preliminary, our results at the Alpysbaev Cave represent one of the earliest dated instances of the Middle Holocene human occupation in the western Tian Shan, especially in the Kazakh portion of the range. New excavations aimed at expanding the existing test pit and uncovering new layers with a larger archaeological assemblage and samples for absolute dating (both OSL and C14) will provide a more detailed chrono-cultural context for human occupation at the Alpysbaev Cave.

## 5.2 The Iron Age population lifeways and activity patterns

Among our sample, there is no evidence of sex bias. Out of the six teeth sampled for aDNA, genetic sex revealed a 1:1 ratio of males and females. A forthcoming genome-wide study will further our understanding by integrating data on biological kinship and population genetics.

The extant skeletal evidence additionally provides a window into the activity patterns of the buried individuals. Constant mechanical stress can result in the hypertrophy of bone at the places where the strained muscles attach [36]. The marked deltoid tuberosity on an otherwise gracile right humerus (TPIT1–40) points to repeated and intense use of the deltoid muscle, potentially during activities requiring arm flexion, rotation, and abduction. Similar patterns of activity-induced skeletal modifications are seen among other archaeological sites and used for evaluating subsistence labor intensity [35,36] or

combat training [49]. The individuals recovered at Alpysbaev Cave likely experienced mechanical stresses consistent with physically demanding lifestyles in Iron Age southern Kazakhstan [50]. This is especially true for farming activities, where muscles of the upper body, arms, and shoulders are actively engaged.

Two instances of rib injuries may suggest interpersonal violence. One typical rib, TPIT1–49, shows a minor healed break on the shaft, and another lower rib (TPIT1–389) was infected secondary to trauma (Fig 5E). In the latter example, the trauma was sustained near the vertebral end, likely due to a crushing back injury. The healing status indicates that the injury this individual sustained was not fatal and that they survived past the proliferative phase and into the remodeling phase. Ribs are typically fractured when force is applied to the ventral aspect of the thoracic cage [51]. Thus, while the events that resulted in the fracture are inconclusive, it could have been the result of direct force caused by blunt force trauma or falling. Due to their location, the fifth to ninth ribs are most commonly fractured in these events, whereas fractures of the first to third rib indicate that the injury resulted from a high kinetic force [51].

Another lumbar vertebra exhibits bilateral spondylolysis (TPIT1–178) (Fig 5F), evidenced by the presence of healed stress fractures located along the lamina between the superior and inferior articular facets (i.e., at the pars interarticularis). The progressive increase of transverse distance between the articular facets of the lumbar spine is a unique human feature, creating a pyramidal configuration of the lumbar spine, which makes lumbar lordosis possible [52]. Bilateral spondylolysis can occur through excessive hyperextension of the lumbar spine, as in particular athletes [53], though hereditary narrowing of the articular facet breadth in lower lumbar vertebrae, or both [41]. Thus, the combination of genetics and repetitive stress from hyperextension of the lower back of any individual can result in spondylolysis [41,52]. Without the anchoring of the posterior half, the anterior half of the vertebra can "slip" forward [41], exacerbating DJD elsewhere in the spine and sacrum. This condition is most often found in adults and is globally attested in the archaeological records of populations known to engage in strenuous labor [54–58].

Pathogen aDNA adds another lens with which we can appreciate disease in our sample. While no maxillae or mandibles were preserved, the presence of multiple taxa associated with periodontal disease or caries (*Parvimonas micra, Campylobacter rectus,* and *Treponema denticola*) may speak to dietary patterns at the site. Diets rich in carbohydrates, especially sucrose and fructose, tend to correlate with the highest frequencies of these diseases [59,60]. Alternatively, these conditions can arise due to a deficiency of vitamin C, protein, or the presence of dental calculus [60]. However, additional targeted studies are necessary to confirm these preliminary findings and to refine our understanding of their implications.

Despite the absence of articulated skeletal remains, our bioarchaeological study suggests that the individuals recovered in Alpysbaev Cave were involved in labour-intensive activities. That most of the remains fall within the Kangju period in southern Kazakhstan, all the data suggest that these individuals were a part of Iron Age agro-pastoral societies. However, the lack of well-contextualized material culture from Alpysbaev Cave (e.g., ceramics, weaponry, tools) hinders further discussion on attributing the people to any specific tribal group.

## 6 Conclusions

In this article, we provide results from our excavations at the newly discovered Alpysbaev Cave site (Kazakhstan). A diverse archaeological assemblage demonstrates that the site was used from the Neolithic to the Iron Age and early Medieval periods in the western Tian Shan. Systematic bioarchaeological analysis revealed critical insights into the lifeways of Iron Age populations, including signs of pathologies and the preservation of the ancient human and oral pathogen DNA. A penetrometer probe yielded the presence of at least 70 cm of sediments, facilitating future excavations. Intact deposits towards the back of Alpysbaev Cave provide substantial potential for uncovering early occupation horizons and exploring possible Neolithic burials in the lower sequences. Further excavations and analysis of the materials are essential to fully reconstruct the occupational history of the site.

## Supporting information

**S1 File. Bioarchaeological analysis.**
(DOCX)

## Acknowledgments

We would like to thank students who actively participated in the excavation of the cave, namely AJ Crawford and Jared Yuryi Shiffert. We would also like to thank the administration of the Sairam-Ögem national park for granting permission to conduct archaeological excavations at the cave.

## Author contributions

**Conceptualization:** Abay Namen, Radu Iovita.

**Data curation:** Abay Namen, Cosimo Posth, Radu Iovita.

**Formal analysis:** Saine C. Hernandez Burgos, Aristeidis Varis, Emily Gaul, Guido Alberto Gnecchi-Ruscone, Maria A. Spyrou, Cosimo Posth, Susanne Lindauer.

**Funding acquisition:** Radu Iovita.

**Investigation:** Abay Namen, Saine C. Hernandez Burgos, Emily Coco, Emily Gaul, Guido Alberto Gnecchi-Ruscone, Maria A. Spyrou, Cosimo Posth, Susanne Lindauer, David Naumann, Zhaken Taimagambetov, Radu Iovita.

**Methodology:** Rachel Kalisher, Maria A. Spyrou, Cosimo Posth, Susanne Lindauer, Scott A. Williams, Zhaken Taimagambetov, Radu Iovita.

**Project administration:** Zhaken Taimagambetov, Radu Iovita.

**Resources:** Cosimo Posth, Zhaken Taimagambetov, Radu Iovita.

**Software:** Emily Coco.

**Supervision:** Abay Namen, Scott A. Williams, Zhaken Taimagambetov, Radu Iovita.

**Validation:** Abay Namen, Radu Iovita.

**Visualization:** Abay Namen, Aristeidis Varis, Emily Coco.

**Writing – original draft:** Abay Namen, Saine C. Hernandez Burgos, Aristeidis Varis.

**Writing – review & editing:** Saine C. Hernandez Burgos, Aristeidis Varis, Emily Coco, Rachel Kalisher, Emily Gaul, Guido Alberto Gnecchi-Ruscone, Maria A. Spyrou, Cosimo Posth, Susanne Lindauer, David Naumann, Scott A. Williams, Zhaken Taimagambetov, Radu Iovita.

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
