## [Decision Letter · Decision Letter 0]

7 May 2025

Dear Dr. Iovita,

Thank you for submitting your manuscript to PLOS ONE. After careful consideration, we feel that it has merit but does not fully meet PLOS ONE’s publication criteria as it currently stands. Therefore, we invite you to submit a revised version of the manuscript that addresses the points raised during the review process.

**All comments must addressed in detail before re-submission.**

We look forward to receiving your revised manuscript.

Kind regards,

Peter F. Biehl, PhD

Academic Editor

PLOS ONE

Journal Requirements:

2. Please include a complete copy of PLOS’ questionnaire on inclusivity in global research in your revised manuscript. Our policy for research in this area aims to improve transparency in the reporting of research performed outside of researchers’ own country or community. The policy applies to researchers who have travelled to a different country to conduct research, research with Indigenous populations or their lands, and research on cultural artefacts. The questionnaire can also be requested at the journal’s discretion for any other submissions, even if these conditions are not met.  Please find more information on the policy and a link to download a blank copy of the questionnaire here: https://journals.plos.org/plosone/s/best-practices-in-research-reporting. Please upload a completed version of your questionnaire as Supporting Information when you resubmit your manuscript.”

3. In your manuscript, please provide additional information regarding the specimens used in your study. Ensure that you have reported human remain specimen numbers and complete repository information, including museum name and geographic location.

For more information on PLOS ONE's requirements for paleontology and archeology research, see https://journals.plos.org/plosone/s/submission-guidelines#loc-paleontology-and-archaeology-research .

“This project has received funding from the European Research Council (ERC) under the European Union's Horizon 2020 research and innovation programme (grant agreement n° 714842; PALAEOSILKROAD project).”

**Additional Editor Comments:**

Your manuscript has now been seen by a referee, whose comments are appended below. You will see from these comments that while the referee find your work of potential interest, they have raised substantial concerns that must be addressed. In light of these comments, we cannot accept the manuscript for publication, but would be interested in considering a revised version that addresses these serious concerns.

We hope you will find the referees' comments useful as you decide how to proceed. Should presentation of further data and analysis allow you to address these criticisms, we would be happy to look at a substantially revised manuscript. However, please bear in mind that we will be reluctant to approach the referees again in the absence of major revisions.

Reviewers' comments:

Reviewer's Responses to Questions

**Comments to the Author**

1. Is the manuscript technically sound, and do the data support the conclusions?

Reviewer #1: Partly

2. Has the statistical analysis been performed appropriately and rigorously?

Reviewer #1: N/A

3. Have the authors made all data underlying the findings in their manuscript fully available?

Reviewer #1: Yes

4. Is the manuscript presented in an intelligible fashion and written in standard English?

Reviewer #1: Yes

Reviewer #1: Kazakhstan

Thank you for the opportunity to review this manuscript detailing recent findings in a cave site in Kazakhstan. While, I am not an expert on this region of the world - I have worked during this time period and with bioarchaeological remains and the majority of my comments are focused on the skeletal analysis. I found the manuscript and findings very interesting, and certainly worthy of publication; however, there are some areas that need revisions. In summary, I see several areas where additional analyses could be undertaken (mostly in sex estimates) and some updating of the MNI seems necessary. And, more clarity overall in terms of the analysis and DDX is warranted. I outline my comments below by section.

Methods

For sex estimation - which method(s) were used? Buikstra and Ubelaker is mostly a summary of various methods - so, the authors should specify which methods were used and cite the original articles.

Results

The clavicles do not represent 3 individuals - they represent at least 2 individuals - since there are two left and one right - it is possible that the right and one of the lefts could be from the same individual - since they are all adult individuals. Unless there is a different reason they are clearly not from the same individual?

Osteophytes of the vertebrae may also be age-related - this should be discussed as a DDX for the observation as well

The rib with the large callus looks like osteomyelitis - this condition can be included as a DDX - particularly with the cloacae

Discussion

may need to update MNI in this section

DJD of the cervical vertebrae is not always associated with use (see comments below and above) - this is also age-related and should be included in the DDX - see:

Adams, B. J., Butler, E., Fuehr, S. M., Olivares‐Pérez, F., & Tamayo, A. S. (2024). Radiographic age estimation based on degenerative changes of vertebrae. Journal of Forensic Sciences, 69(2), 391-399.

Pronounced deltoid tuberosity is discussed as being related to male sex and activity - both should be explored together - and not independently. And, may have implications for sex differences in activity - also see note for SI - sex estimation can be made on the humerus.

For the rib - I would hesitate to speculate about pain - pain is a very individualized experience and it is not knowable if they were in joint pain - likely pain due to the infection - but it is not knowable how the joints were impacted by this infection or how pain was experienced

Periodontal disease is in the discussion - but, I don't see any discussion of periodontal disease in the results. In fact the results say that no vira were found - and does not clearly state if bacteria were found

Does genetic analysis increase the MNI to 6? Is this due to different genomic signatures? I only saw discussion of sex estimates in the results, so that would only be 2 individuals? Unless these were all from the same tooth? Then, just the duplicating teeth could provide the MNI. In fact, table 5 should include the tooth for each sex estimate as a column.

Supplemental information

the authors use the word vertebrae when referring to a single vertebra - just an editorial comment to change it to vertebra for the singular

Also, could the authors use methods of centrum age estimation for these individuals with vertebrae? Such as:

Albert, Arlene Midori, and William R. Maples. "Stages of epiphyseal union for thoracic and lumbar vertebral centra as a method of age determination for teenage and young adult skeletons." Journal of forensic sciences 40, no. 4 (1995): 623-633

and other related methods?

spondylolysis - is related to activity and genetics as well - being well healed is part of spondylolysis - so, this statement can be included with the DDX as part of the evidence for spondylolysis

The right humerus could also have a sex estimate following the method of:

Rogers, Tracy L. "A visual method of determining the sex of skeletal remains using the distal humerus." Journal of Forensic Sciences 44.1 (1999): 57-60.

This could serve as way to verify the observation of robusticity

Also, based on the vertical diameter of the humeral head:

Stewart has methods for metrics of the humerus, as does:

Spradley, M. Katherine, and Richard L. Jantz. "Sex estimation in forensic anthropology: skull versus postcranial elements." Journal of forensic sciences 56, no. 2 (2011): 289-296.

Can more detail be given of the perinate tibia? A photo of the nutrient foramen would be helpful. From the photos provided, it does not look like perinate tibia

Were the discussed cranial fragments not studied? There is discussion of cranial fragments in the text, but they are not in the SI? Nor are they included in table 2 of the main document

**Do you want your identity to be public for this peer review?** For information about this choice, including consent withdrawal, please see our Privacy Policy

Reviewer #1: No

---

## [Author Response · Author response to Decision Letter 1]

23 Jul 2025

Dear Dr. Biehl,

Thank you for the opportunity to revise our manuscript. Below, we provide detailed responses to each of the comments, along with a summary of the changes made to the manuscript.

REVIEWER 1:

Thank you for the opportunity to review this manuscript detailing recent findings in a cave site in Kazakhstan. While, I am not an expert on this region of the world - I have worked during this time period and with bioarchaeological remains and the majority of my comments are focused on the skeletal analysis. I found the manuscript and findings very interesting, and certainly worthy of publication; however, there are some areas that need revisions. In summary, I see several areas where additional analyses could be undertaken (mostly in sex estimates) and some updating of the MNI seems necessary. And, more clarity overall in terms of the analysis and DDX is warranted. I outline my comments below by section.

Methods

For sex estimation - which method(s) were used? Buikstra and Ubelaker is mostly a summary of various methods - so, the authors should specify which methods were used and cite the original articles.

We agree and have now clarified the specific methods used for sex estimation. We now cite Klales et al. 2012 and Walker 2005 on revised MS lines 118-120.

Results

The clavicles do not represent 3 individuals - they represent at least 2 individuals - since there are two left and one right - it is possible that the right and one of the lefts could be from the same individual - since they are all adult individuals. Unless there is a different reason they are clearly not from the same individual?

Yes, thank you for this comment. We have revised the discussion of the clavicles and changed the text to indicate that they represented two young adult individuals. Please see revised MS lines 240-246.

Osteophytes of the vertebrae may also be age-related - this should be discussed as a DDX for the observation as well.

We have added mention of this to the manuscript on revised MS lines 267-269.

The rib with the large callus looks like osteomyelitis - this condition can be included as a DDX - particularly with the cloacae.

We have added mention of osteomyelitis to the manuscript on revised MS lines 318-319.

Discussion

may need to update MNI in this section

We have updated and clarified the MNI in the discussion sections of the manuscript.

DJD of the cervical vertebrae is not always associated with use (see comments below and above) - this is also age-related and should be included in the DDX - see:

Adams, B. J., Butler, E., Fuehr, S. M., Olivares‐Pérez, F., & Tamayo, A. S. (2024). Radiographic age estimation based on degenerative changes of vertebrae. Journal of Forensic Sciences, 69(2), 391-399.

See above.

Pronounced deltoid tuberosity is discussed as being related to male sex and activity - both should be explored together - and not independently. And, may have implications for sex differences in activity - also see note for SI - sex estimation can be made on the humerus.

We adjusted our discussion of the deltoid tuberosity in revised MS lines 447-451 and no longer link the pronounced tuberosity to a male sex estimate.

For the rib - I would hesitate to speculate about pain - pain is a very individualized experience and it is not knowable if they were in joint pain - likely pain due to the infection - but it is not knowable how the joints were impacted by this infection or how pain was experienced

Thank you, we have removed explicit mention of pain.

Periodontal disease is in the discussion - but, I don't see any discussion of periodontal disease in the results. In fact the results say that no vira were found - and does not clearly state if bacteria were found

We now explicitly acknowledge that PD was not observed in maxillae or mandibles, but only through the presence of pathogens associated with PD. Please see revised MS lines 479-482.

Does genetic analysis increase the MNI to 6? Is this due to different genomic signatures? I only saw discussion of sex estimates in the results, so that would only be 2 individuals? Unless these were all from the same tooth? Then, just the duplicating teeth could provide the MNI.

We do not estimate the MNI based on the produced DNA data. Therefore, the genetic sex estimates do not contribute to the MNI inferred from the other analyses.

In fact, table 5 should include the tooth for each sex estimate as a column.

We agree and have added that information to the table.

Supplemental information

The authors use the word vertebrae when referring to a single vertebra - just an editorial comment to change it to vertebra for the singular

Thank you for catching that spelling error. We have amended all instances of this.

Also, could the authors use methods of centrum age estimation for these individuals with vertebrae? Such as:

Albert, Arlene Midori, and William R. Maples. "Stages of epiphyseal union for thoracic and lumbar vertebral centra as a method of age determination for teenage and young adult skeletons." Journal of forensic sciences 40, no. 4 (1995): 623-633

and other related methods?

We use apophyseal fusion as a means to estimate age in the vertebrae.

spondylolysis - is related to activity and genetics as well - being well healed is part of spondylolysis - so, this statement can be included with the DDX as part of the evidence for spondylolysis

Thank you for this comment. We have significantly expanded our discussion on spondylolysis to include explicit mention of its epigenetic nature. Please see revised MS lines 466-478.

The right humerus could also have a sex estimate following the method of:

Rogers, Tracy L. "A visual method of determining the sex of skeletal remains using the distal humerus." Journal of Forensic Sciences 44.1 (1999): 57-60.

This could serve as way to verify the observation of robusticity

Also, based on the vertical diameter of the humeral head:

Stewart has methods for metrics of the humerus, as does:

Spradley, M. Katherine, and Richard L. Jantz. "Sex estimation in forensic anthropology: skull versus postcranial elements." Journal of forensic sciences 56, no. 2 (2011): 289-296.

Thank you for this suggestion, however without a larger sample within which we can seriate, we do not feel comfortable estimating sex from the single humerus.

Can more detail be given of the perinate tibia? A photo of the nutrient foramen would be helpful. From the photos provided, it does not look like perinate tibia

We have provided more information in the supplemental text, and now express caution in the perinatal tibia estimate for the very reason you identify. The nutrient foramen is difficult to appreciate in the provided photos, but in the 3D model, the morphology is not entirely consistent with a perinatal tibia.

Were the discussed cranial fragments not studied? There is discussion of cranial fragments in the text, but they are not in the SI? Nor are they included in table 2 of the main document

The cranial fragments were not studied as they were not excavated, but we have now amended the text to make this

---

## [Decision Letter · Decision Letter 1]

18 Sep 2025

Human occupations at the Alpysbaev Cave (western Tian Shan): Bioarchaeological insights from the Iron Age burial cluster

PONE-D-25-16735R1

Dear Dr. Iovita,

We’re pleased to inform you that your manuscript has been judged scientifically suitable for publication and will be formally accepted for publication once it meets all outstanding technical requirements.

Kind regards,

Peter F. Biehl, PhD

Academic Editor

PLOS ONE

Additional Editor Comments (optional):

Reviewer #1:

Reviewers' comments:

Reviewer's Responses to Questions

**Comments to the Author**

Reviewer #1: All comments have been addressed

2. Is the manuscript technically sound, and do the data support the conclusions?

Reviewer #1: (No Response)

3. Has the statistical analysis been performed appropriately and rigorously?

Reviewer #1: (No Response)

4. Have the authors made all data underlying the findings in their manuscript fully available?

Reviewer #1: (No Response)

5. Is the manuscript presented in an intelligible fashion and written in standard English?

Reviewer #1: (No Response)

Reviewer #1: Thank you for the thoughtful revision and consideration of comments. The manuscript is now suitable for publication with no further revisions.

**Do you want your identity to be public for this peer review?** For information about this choice, including consent withdrawal, please see our Privacy Policy

Reviewer #1: No

---

## [Editor Report · Acceptance letter]

PONE-D-25-16735R1

PLOS ONE

Dear Dr. Iovita,

I'm pleased to inform you that your manuscript has been deemed suitable for publication in PLOS ONE. Congratulations! Your manuscript is now being handed over to our production team.

Kind regards,

on behalf of

Dr. Peter F. Biehl

Academic Editor

PLOS ONE